# Alpha-Glycerylphosphorylcholine Increases Motivation in Healthy Volunteers: A Single-Blind, Randomized, Placebo-Controlled Human Study

**DOI:** 10.3390/nu13062091

**Published:** 2021-06-18

**Authors:** Yasuhisa Tamura, Kumi Takata, Kiminori Matsubara, Yosky Kataoka

**Affiliations:** 1RIKEN Center for Biosystems Dynamics Research, Laboratory for Cellular Function Imaging, 6-7-3 Minatojima-Minamimachi, Chuo-ku, Kobe 650-0047, Japan; kumi@riken.jp (K.T.); kataokay@riken.jp (Y.K.); 2RIKEN-JEOL Collaboration Center, Multi-Modal Microstructure Analysis Unit, 6-7-3 Minatojima-Minamimachi, Chuo-ku, Kobe 650-0047, Japan; 3Graduate School of Humanities and Social Sciences, Hiroshima University, 1-1-1 Kagamiyama, Higashi-Hiroshima City, Hiroshima 739-8524, Japan; kmatsuba@hiroshima-u.ac.jp

**Keywords:** alpha-glycerylphosphorylcholine, motivation, monoamine, placebo, KOKORO scale

## Abstract

Alpha-glycerylphosphorylcholine (αGPC) is a precursor of acetylcholine and can increase acetylcholine concentration in the brain. In addition, αGPC has a role in cholinergic function as well as monoaminergic transmission, including dopaminergic and serotonergic systems. These monoaminergic systems are related to feelings and emotions, including motivation, reward processing, anxiety, and depression. However, the precise effects of αGPC on human feelings and emotions remain to be elucidated. In this study, we investigated changes in the subjective feelings of healthy volunteers using the KOKORO scale before and after administering αGPC. Thirty-nine volunteers participated in a single-blind, placebo-controlled design. Participants completed a KOKORO scale test to quantify self-reported emotional states, three times each day for two weeks preceding treatment and then for a further two weeks while self-administering treatment. αGPC treatment show a tendency to increase motivation during the intervention period. Furthermore, motivation at night was significantly higher in the αGPC group than in the placebo group (*p* < 0.05). However, αGPC did not show any effects on anxiety. These data suggest that αGPC can be used to increase motivation in healthy individuals.

## 1. Introduction

Alpha-glycerylphosphorylcholine (αGPC) is a deacylated phosphatidylcholine derivative and is widely used as a dietary supplement. In addition, αGPC is a precursor of acetylcholine and can enhance cholinergic transmission in the brain [1,2]. Pre-clinical studies have shown that αGPC has an anti-amnesic effect against scopolamine-induced amnesia [1] and a neuroprotective effect in animal models of cerebrovascular disease [3]. Moreover, αGPC ameliorates seizure-induced cognitive impairment by reducing neuronal cell death and blood–brain barrier disruption [4]. Clinical trials have been performed to evaluate the efficacy of αGPC for treating cognitive decline in patients with Alzheimer’s disease [5,6] and vascular dementia [7,8,9]. Treatment with αGPC increases dopamine levels as well as dopamine active transporter expression in the frontal cortex and cerebellum [10]. It also increases serotonin levels in the frontal cortex and striatum of rat brains [10]. These data suggest that αGPC has the potential to act as a modulator of the dopaminergic (DA) neurotransmission and serotonergic (5-HT) system in addition to affecting cholinergic circuits. DA neurons are located in three distinct nuclei: ventral tegmental area (VTA), substantia nigra, and retrorubral field of the brain. Among them, DA neurons in the VTA project to the ventromedial striatum (nucleus accumbens) [11] and the DA transmission in nucleus accumbens has a crucial role in motivational processes [12,13,14]. Changes in DA neurotransmission are linked to regulation of motivated behavior [15,16]. On the other hand, 5-HT neurons are arranged in the brainstem and their cell bodies are mainly present in raphe nucleus [17]. 5-HT receptors are composed of 7 subtypes (5-HT1-7) and are associated with anxiety and depression [18]. Indeed, 5-HT1A receptor knockout mice show increased anxiety-like behavior [19]. These findings suggest that αGPC may affect motivation and anxiety in healthy volunteers. However, to date, there are no reports on the effects of αGPC on human feelings and emotions. We have previously demonstrated the successful quantification of emotional states, including anxiety, tension, security, relaxation, and refreshment, using the KOKORO scale [20]. The KOKORO scale consists of a four-quadrant matrix in a two-dimensional space and can easily record slight changes in subjective feelings and emotions with smartphones. We investigated the effects of αGPC on anxiety–relief and lethargy–motivation in healthy volunteers using the KOKORO scale.

## 2. Materials and Methods

### 2.1. Subjects

The study was conducted in accordance with the Declaration of Helsinki and was approved by the Ethics Committee of the RIKEN Center for Biosystems Dynamics Research (Kobe2 2018-02) (19 June 2018). Forty volunteers (31 females, 9 males) aged 22–59 years were recruited between July and October of 2018 for this study. The inclusion criterion was healthy individuals aged between 20 and 60 years, and exclusion criteria included: (1) women who are planning pregnancy, currently pregnant, or lactating, (2) individuals visiting the hospital and/or were on medications for diseases, and (3) smokers. All participants provided written informed consent before beginning the present study. The study was registered with the University hospital Medical Information Network (UMIN) Clinical Registry (UMIN000044563).

Participants were randomly divided into placebo and αGPC treatment groups by age and sex. One participant in the placebo group was excluded from data analysis for lack of compliance with data entry in the KOKORO scale system (≤75%). Thus, KOKORO data from 19 participants in the placebo group (age 43.6 ± 9.4 years; 15 females and 4 males) and 20 in the αGPC group (age 43.1 ± 8.2 years; 16 females and 4 males) were used for analysis (Figure 1).

### 2.2. Treatment

Participants were administered capsules containing either 200 mg of αGPC, which was purchased from NOF Corporation (NICHIYU^®^ GPC85R, powder containing 85% αGPC) (Tokyo, Japan), or cellulose (placebo). The participants self-administered two capsules, once daily at bedtime for 2 weeks, for a total daily dose of 400 mg αGPC in the treatment group. Previous clinical studies have shown that there are no serious side effects or toxicities when human subjects were orally administered αGPC (1200 mg/day) for 6 months [5,6]. In animal studies, LD_50_ values of αGPC in rodent and dogs were estimated at ≥10 and ≥3 g/kg respectively, by oral administration [21]. Prolonged administration of αGPC (rodents: ≥1000 mg/kg/day, dogs: ≥300 mg/kg/day) hardly show serious adverse events for 26 weeks [21]. Thus, the doses of αGPC and the duration of treatment in this study were determined based on these previous reports.

### 2.3. Assessment of Human Feelings and Emotions Using the KOKORO Scale

The KOKORO scale was used to quantify and monitor changes in human feelings and emotions: anxiety and motivation were evaluated using the four-quadrant matrix in a two-dimensional field, as shown in Figure 2A [20]. In this study, the x-axis and y-axis of the KOKORO scale were set to “Anxiety–Relief” and “Lethargy–Motivation,” respectively (Figure 2A). The participants were instructed to input their feelings into the four-quadrant chart on their smartphones thrice daily for two weeks. The coordinate data were treated as measured values from −100 to 100 (continuous values). The delta (post–pre) values were calculated by subtracting the measured values of pre- and post-intervention each week (first and second week) (Figure 2B).

### 2.4. Study Design

A single-blind, randomized, placebo-controlled design was used in this study. The participants were instructed to record their feelings using the KOKORO scale on their smartphone screens three times a day for four weeks and to consume two capsules of αGPC or placebo, as described above, immediately after data input at night during those two weeks (Figure 2B). Daily entry was performed after waking up in the morning (6:00–11:00, morning), after lunch (12:00–17:00, afternoon), and before bedtime (19:00–00:00, night).

### 2.5. Statistical Analysis

Analyses were performed using the GraphPad Prism software (v6.0; GraphPad Software Inc., San Diego, CA, USA). Statistically significant differences between the two groups were determined using the Mann–Whitney *u* test with Bonferroni multiple comparison corrections. A value of *p* < 0.05 was considered to represent a statistically significant result.

## 3. Results

In this study, subject recruitment started in July 2018 and was completed in October 2018. No serious side effects were observed during the clinical trials. Changes in subjective feelings in both placebo and αGPC groups were calculated as the difference between values before and after treatment. As shown in Figure 3A, both placebo and αGPC groups experienced relief while receiving capsules for 2 weeks. However, comparison between groups showed no significant differences. The effects of placebo and αGPC capsules on motivation in healthy volunteers are shown in Figure 3B. The placebo had no effect on motivation, whereas the αGPC group showed an increasing trend in motivation during the intervention period. We investigated the changes in feelings of both groups within each time block (morning, afternoon, and night). In the anxiety–relief axis, the change in feelings in both groups showed similar patterns across all time blocks (Figure 3C), whereas in the lethargy–motivation axis, the change in feelings in the αGPC group increased significantly at night when compared with that in the placebo group (Figure 3D). 

## 4. Discussion

αGPC is a water-soluble deacylated metabolite of phosphatidylcholine and a precursor of acetylcholine. In clinical trials, αGPC improved cognitive impairment in patients with Alzheimer’s disease [5,6] and vascular dementia [7,8,9]. Furthermore, αGPC ameliorated cognitive decline in several experimental animal models [1,4,22,23,24] and enhanced acetylcholine levels in the brains of rodents [1,22,23]. These reports suggest that αGPC might show ameliorating effects on cognitive function via increased cholinergic transmission. In addition, αGPC increases the level of DA and 5-HT and the expression of DAT in rat brains [10]. The report indicates that αGPC functions as a modulator of DA and 5-HT systems as well as cholinergic transmission. DA transmission is closely linked to motivation and reward processing [12,13,14], whereas the 5-HT system is associated with anxiety and depression [18,19]. These findings provoked the hypothesis that αGPC affects human feelings and emotions, including motivation, anxiety, and depression. In this study, this hypothesis was investigated with healthy volunteers. The administration of αGPC showed a tendency to increase self-reported motivation levels during the intervention period (Figure 3B), and in particular, significantly improved those at night (Figure 3D). In a previous study with rats, αGPC improved cognitive decline following seizure, and such effects were observed from 3 weeks, but were not found at 1 week of αGPC administration [4]. These data suggest that long-term administration (>1 week) is necessary for the onset of action of αGPC. Moreover, αGPC shows blood–brain barrier permeability and can enter neuronal cell bodies of rodent brains when orally administered [25]. Therefore, αGPC may affect feelings of increased motivation in human subjects, potentially through interaction with dopamine circuits in the brain. However, their mechanism of action remains unclear at present. αGPC did not show specific effects on anxiety (Figure 3A,C). However, the placebo group showed an increased feeling of relief compared with the αGPC treatment group (Figure 3A), which may be the result of placebo. Indeed, it has been reported that placebo can have anti-anxiety effects in humans [26,27].

In this clinical trial, healthy subjects self-administered a total αGPC dosage of 400 mg per day for 2 weeks. This dose of αGPC was determined based on previous studies in animals and humans. In dogs, chronic administration of αGPC (≥300 mg/kg/day) did not show serious adverse effects for 26 weeks [21]. In addition, clinical trials showed that there are no severe side effects or toxicities following oral administration of αGPC (1200 mg/day) for 6 months. These findings suggested that the risk of toxicity in administering 400 mg/day of αGPC for 2 weeks was extremely low. Indeed, we confirmed that αGPC (400 mg/day) administration did not exhibit any side effects or addictions during the 2 weeks of intervention and at least 12 months after taking the medication. Nevertheless, we showed that αGPC exhibited a tendency to increase motivation during the intervention period.

However, the present study has some limitations. In this study, we could not address whether αGPC acted directly or indirectly on human motivation and neural circuits, especially the dopamine system in the human brain. Further research with experimental animals may resolve this limitation. Next, this study was conducted on a relatively small scale, using a short-term (2 weeks) intervention and a single-blind test as a pilot clinical trial. Thus, our results can be validated by performing further studies on a large scale and with long-term intervention. In this study, the investigation and the data analysis were performed by different researchers to minimize experimental biases. Finally, the changes of human feelings and emotions might be affected by several factors and events in daily life. Thus, the participants were instructed to input the matters with their feelings if they experienced extraordinary events and sensations. These data were excluded from the data analysis.

## 5. Conclusions

We showed that use of αGPC increased motivation in human subjects without serious side effects in this study. These findings suggest that αGPC supplementation has an increased effect on motivation of healthy subjects.

## Figures and Tables

**Figure 1 nutrients-13-02091-f001:**
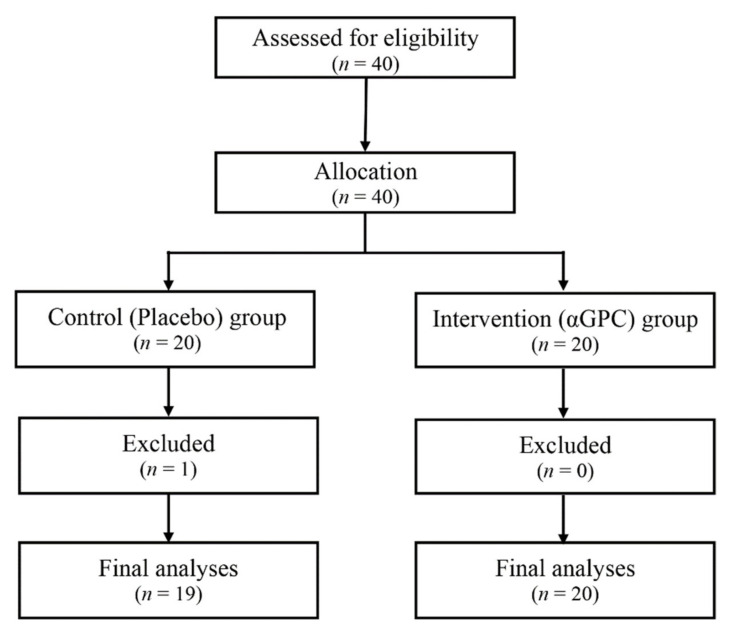
Flow diagram of patient recruitment in this study.

**Figure 2 nutrients-13-02091-f002:**
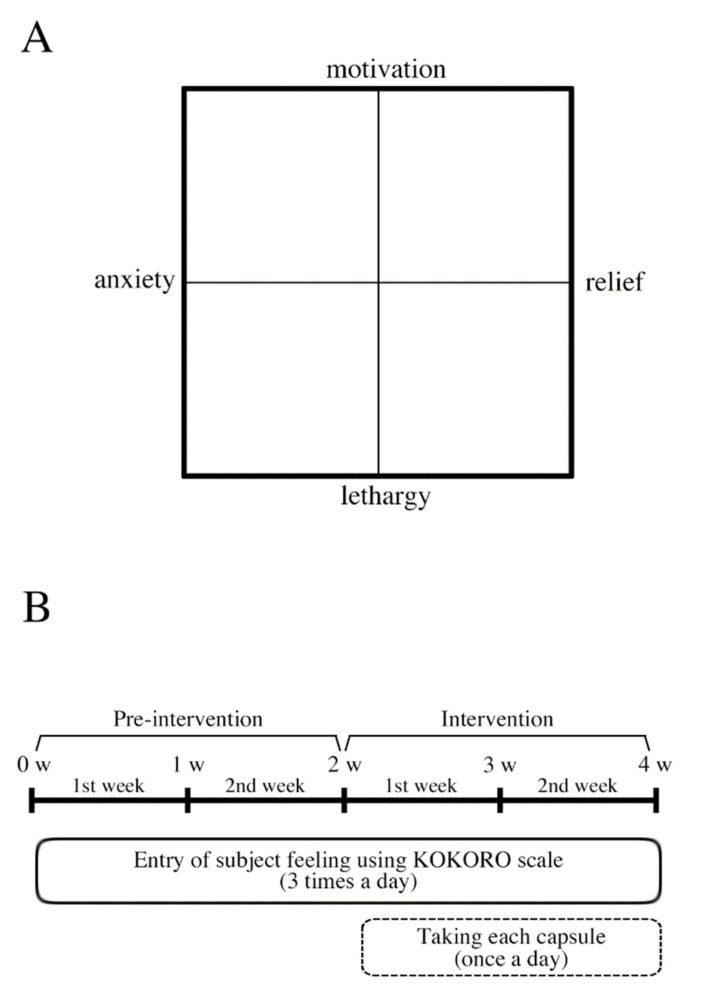
Schematic illustration of KOKORO scale and the experimental design in a human study. (**A**) KOKORO scale for evaluating human subjective feelings. (**B**) Each participant entered his/her subjective feelings using the KOKORO scale three times (morning, afternoon, and night) each day for four weeks (solid line box) and self-administered the treatment they were assigned (before bedtime) for two weeks (dashed line box). Pre-intervention and intervention show the period without and with the assigned capsule intake, respectively.

**Figure 3 nutrients-13-02091-f003:**
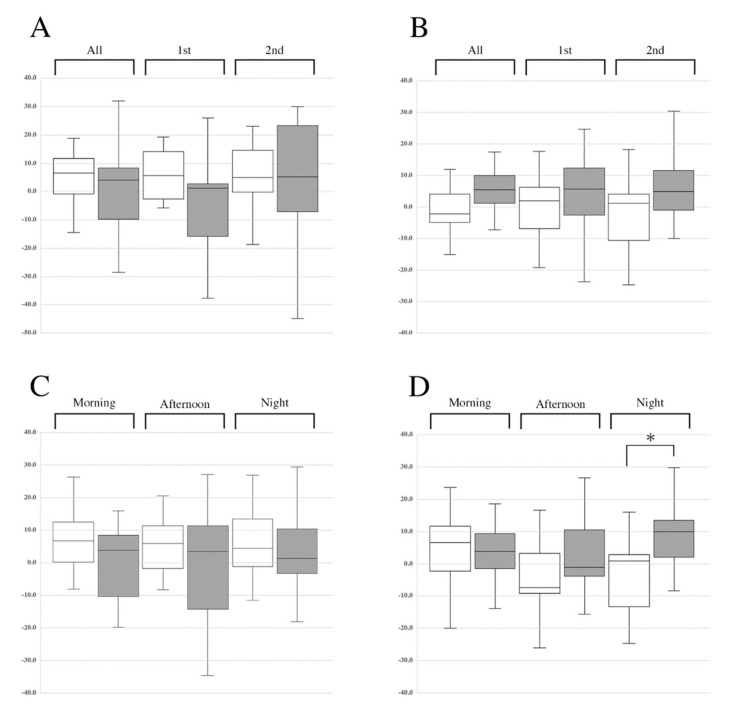
Alteration of subjective feelings in healthy volunteers before and after taking αGPC. Box and whisker plots for changes in anxiety–relief (**A**,**C**) and lethargy–motivation (**B**,**D**) in placebo (white bars) and αGPC (colored bars) groups. (**A**,**B**) Dosing period, All: entire period (1st and 2nd week), 1st: first week, 2nd: second week. (**C**,**D**) Time blocks (morning, afternoon, and night). Boxes represent the first and third quartiles. Whiskers represent the range from minimum to maximum of observed data. * *p* < 0.05.

## Data Availability

The data presented in this study are available on request from the corresponding author.

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
