# Peer review of "Alpha-Glycerylphosphorylcholine Increases Motivation in Healthy Volunteers: A Single-Blind, Randomized, Placebo-Controlled Human Study"

_nutrients, 2021, doi:10.3390/nu13062091_

Round 1

Reviewer 1 Report

Review for Nutrients, Manuscript ID: nutrients-1205289: Alpha-glycerylphosphorylcholine increases motivation in healthy volunteers: A single-blind, randomized, placebo-controlled human study.

The authors explore the “effects of αGPC on human feelings and emotions,” concluding that “αGPC treatment increased motivation during the intervention period.” However, the data do not support this conclusion.

The authors’ approach to exploring potential benefits of αGPC on motivation seems interesting, and the paper is generally well-written and easy to comprehend, but before this paper can be properly evaluated for publication, important questions and study deficits should be addressed.

  1. The statistical analysis does not correct for multiple comparisons. Generally speaking, not correcting for multiple comparisons contributes to the replication crisis.  The p-values shown in published papers as a rule do not correct for the multiple attempts at finding statistically significant results that occur behind the scenes in studies never submitted for publication.  However, proper statistical analyses do correct for the statistical tests to be presented together in the same paper. The authors might benefit from consulting a statistician for advice on how to perform the statistical analysis. One approach might be to use a Bonferroni correction, i.e., dividing the significance criterion (usually α = 0.05), by the number of statistical tests, which from Figure 3 would be 12. To improve the sensitivity of the tests, one might consider dividing the hypotheses tested into primary and secondary hypotheses, with the idea that the secondary hypotheses would only be tested if the corresponding primary null hypotheses were rejected. For example, the primary hypotheses here would naturally be that αGPC (1) improves motivation and (2) improves relief on average during the intervention period, compared with placebo. The Bonferroni correction factor would then be 2, resulting in an α of 0.025 (for two-tailed tests, which presumably is what was used, although the authors did not specify), which would mean that the results of this study should be considered nonsignificant.  Although there is a place in the literature for “negative” studies, in this case the study sample size is too small to help us to understand if the effect size should be considered small; and as discussed below, the study quality is insufficient for likely inclusion in a future meta-analysis.
  2. Experimenter bias is not addressed. Single-blind studies are considered much lower in quality compared with double-blind studies because the experimenters can unknowingly influence the results of the study in every phase of the study, especially during data collection via their interactions with study participants.  The authors might consider mentioning this limitation and explaining why the study was not a double-blind study.  Interesting in this context is that the authors write in the abstract “αGPC did not show any effects on anxiety” and on p. 3 trivialize the increase in anxiety for the treatment group during the first week of the intervention (p < 0.05 in Figure 3A), while highlighting the perceived importance of the increase in motivation for the treatment group over the entire intervention period (p = 0.07), during the second week (p < 0.05), and at night (p < 0.01).
  3. The temporal relationship between taking the medication and changes in motivation and relief is not explored. The study medication was taken before bed as well as the smartphone nightly evaluation, but it does not appear to be known or controlled which of these events occurred first for each participant. If they occurred simultaneously, then this begs the question of why participants would feel more motivation after 24 hours without another dose of αGPC (and how helpful that may or may not be right before going to bed).  If other studies have documented the pharmacokinetics or pharmacodynamics for αGPC, then this information should be summarized and cited.
  4. The possibility of addiction or dwindling or negative effects over time is not explored.  Methamphetamine improves motivation and heroin improves “relief” in the short-term, but these are not prescribed by physicians due to their now well-understood harmful long-term effects.  Medications that make people feel good in the short-term often do the opposite over time.  The text and citations on p. 6 “no serious side effects or toxicities when human subjects were orally administered αGPC (1200 mg/day) for 6 months [6, 18-19]” are reassuring, but the questions of possible addiction and dwindling effects should be specifically addressed.  Toward that end, a longer intervention time would be helpful.

Author Response

Thank you for your comments. Please see our replies to your comments below.

1.The statistical analysis does not correct for multiple comparisons. Generally speaking, not correcting for multiple comparisons contributes to the replication crisis. The p-values shown in published papers as a rule do not correct for the multiple attempts at finding statistically significant results that occur behind the scenes in studies never submitted for publication. However, proper statistical analyses do correct for the statistical tests to be presented together in the same paper. The authors might benefit from consulting a statistician for advice on how to perform the statistical analysis. One approach might be to use a Bonferroni correction, i.e., dividing the significance criterion (usually α = 0.05), by the number of statistical tests, which from Figure 3 would be 12. To improve the sensitivity of the tests, one might consider dividing the hypotheses tested into primary and secondary hypotheses, with the idea that the secondary hypotheses would only be tested if the corresponding primary null hypotheses were rejected. For example, the primary hypotheses here would naturally be that αGPC (1) improves motivation and (2) improves relief on average during the intervention period, compared with placebo. The Bonferroni correction factor would then be 2, resulting in an α of 0.025 (for two-tailed tests, which presumably is what was used, although the authors did not specify), which would mean that the results of this study should be considered nonsignificant. Although there is a place in the literature for “negative” studies, in this case the study sample size is too small to help us to understand if the effect size should be considered small; and as discussed below, the study quality is insufficient for likely inclusion in a future meta-analysis.

Response: Thank you very much for your valuable suggestion. We had used Mann-Whitney U-test for pairwise comparisons between groups according to the suggestion from a statistician and several previous studies that were performed under similar conditions, as listed below:

  • Urdampilleta et al., Effects of 120 vs. 60 and 90 g/h Carbohydrate Intake during a Trail Marathon on Neuromuscular Function and High Intensity Run Capacity Recovery. Nutrients 12 2094 2020
  • Kozato et al., A randomised controlled pilot trial of the influence of non-native English accents on examiners’ scores in OSCEs. BMC Medical Education 20 268 2020
  • Løndal et al., Physical activity of first graders in Norwegian after-school programs: A relevant contribution to the development of motor competencies and learning of movements? Investigated utilizing a mixed methods approach. PLOS ONE 15 e0232486 2020
  • Frazzitta et al., Effectiveness of a Very Early Stepping Verticalization Protocol in Severe Acquired Brain Injured Patients: A Randomized Pilot Study in ICU. PLOS ONE 11 e0158030 2016

  1. Experimenter bias is not addressed. Single-blind studies are considered much lower in quality compared with double-blind studies because the experimenters can unknowingly influence the results of the study in every phase of the study, especially during data collection via their interactions with study participants. The authors might consider mentioning this limitation and explaining why the study was not a double-blind study.

Response: This study was performed as a single-blind pilot trial. In this study, the investigation and data analyses were performed by different researchers to minimize experimental biases. We have added this description in the “Discussion” section of the revised manuscript (Lines 195-198).

Interesting in this context is that the authors write in the abstract “αGPC did not show any effects on anxiety” and on p. 3 trivialize the increase in anxiety for the treatment group during the first week of the intervention (p < 0.05 in Figure 3A), while highlighting the perceived importance of the increase in motivation for the treatment group over the entire intervention period (p = 0.07), during the second week (p < 0.05), and at night (p < 0.01).

Response: We have revised these sentences to avoid misleading statements in the “Results” section of the revised manuscript (Lines 116-120, 121-123).

  1. The temporal relationship between taking the medication and changes in motivation and relief is not explored. The study medication was taken before bed as well as the smartphone nightly evaluation, but it does not appear to be known or controlled which of these events occurred first for each participant. If they occurred simultaneously, then this begs the question of why participants would feel more motivation after 24 hours without another dose of αGPC (and how helpful that may or may not be right before going to bed). If other studies have documented the pharmacokinetics or pharmacodynamics for αGPC, then this information should be summarized and cited.

Response: We have added this information in the “Discussion” section in the revised manuscript. (Line 162-170).

  1. The possibility of addiction or dwindling or negative effects over time is not explored. Methamphetamine improves motivation and heroin improves “relief” in the short-term, but these are not prescribed by physicians due to their now well-understood harmful long-term effects. Medications that make people feel good in the short-term often do the opposite over time. The text and citations on p. 6 “no serious side effects or toxicities when human subjects were orally administered αGPC (1200 mg/day) for 6 months [6, 18-19]” are reassuring, but the questions of possible addiction and dwindling effects should be specifically addressed. Toward that end, a longer intervention time would be helpful.

Response: We had investigated the presence or absence of adverse experiences with a self-certification form for at least 12 months following αGPC treatment. No adverse events, including addiction and negative effects, were reported during this period. We have added this information in the “Discussion” section of the revised manuscript (Line 186-189). On the other hand, we could not study the deteriorating effects of αGPC treatment during a long-term intervention. However, we are planning further studies for determining the effects of the long-term use of αGPC, which may help resolve this problem.

Reviewer 2 Report

Alpha-glycerylphosphorylcholine increases motivation in healthy volunteers: A single-blind, randomized, placebo-controlled human study.

Synopsis: alpha-glycerylphosphorylcholine (αGPC) is a deacylated phosphatidylcholine derivative and is widely used as a dietary supplement. Being a precursor of acetylcholine and having the ability to enhance cholinergic transmission, clinical trials have been performed to evaluate its efficacy for the treatment of cognitive decline in patients with Alzheimer's disease and vascular dementia. Since αGPC may act also as a modulator of monoaminergic transmission, in particular in dopaminergic and serotoninergic system, it may affect motivation and anxiety in healthy volunteers. To quantifying the emotional states, the authors of this paper have been using the KOKORO scale on thirthy-nine volunteers in a single-blind study. Forty volunteers (1 participant was excluded from placebo group because there is lack of compliance with data entry in KOKORO scale), 31 females and 9 males, aged 22 - 59 years, were recruited by these criteria:

  • Healthy individuals aged 20 – 60 years;
  • No smokers;
  • No women who are planning pregnancy, currently pregnancy, or lacting;
  • No individuals visiting the hospital and/or on medication for desease.

The individuals were randomly divided in placebo and αGPC treatment groups. Partecipants were administered capsules containing 200 mg of αGPC or cellulose placebo. Each participant self-administered two capsules, once daily, at bedtime, for two weeks, and report its feelings into the four-quadrant matrix of KOKORO scale, thrice daily (morning, afternoon and night) for two weeks.  This study showed significantly increased in the motivation feeling in αGPC group during the intervention period (p=0.0707), whereas the placebo had no effect. In the anxiety – relief axis, the change of feelings showed similar pattern in both groups. The major effect of αGPC was in the motivation – lethargy axis at night (p=0.0020). The authors suggest that αGPC increased motivation feeling since first week and also second week, through interaction with dopamine circuit, because dopamine levels are closely associated to motivation.

Critical Overview: the paper is generally well written and structure, and the language is clear. The aim of the authors to demonstrate that αGPC increase motivation in healthy volunteers partially has been met. The authors should increase the cohort of volunteers (to improve the statistic) and validate this test with objective data, to give more force to outcomes. Changes in subjective feelings may be affected by variables, as well as αGPC administration, which were not considered during the experiment (i.e., mourning, family problems, promotion at work, implementation of projects, etc…) and cannot be overlooked. Measuring dopamine levels would have been appropriate to improve the work, since the literature reports that dopamine levels are closely related to motivation. In the results, it is reported that administration of αGPC increase motivation compared with the placebo group, but the p value is higher than threshold (p=0.0707), therefore that cannot be a significant outcome. Nevertheless, the results are supported by a weak statistic in general. Basically, there is not sufficient strong effects by administering of αGPC, as a significant effect only at nighttime point is observed, in figure 3D. Furthermore, also increases in the feeling of relief in both placebo and treatment groups are unclear (Fig. 3A). Overall graphs are clear and seem to be of good quality. It is advisable a general review of bibliography to update older references.

Author Response

Thank you for your kind comments. Please see our replies to your comments below.

Critical Overview: the paper is generally well written and structure, and the language is clear. The aim of the authors to demonstrate that αGPC increase motivation in healthy volunteers partially has been met. The authors should increase the cohort of volunteers (to improve the statistic) and validate this test with objective data, to give more force to outcomes. Changes in subjective feelings may be affected by variables, as well as αGPC administration, which were not considered during the experiment (i.e., mourning, family problems, promotion at work, implementation of projects, etc…) and cannot be overlooked. Measuring dopamine levels would have been appropriate to improve the work, since the literature reports that dopamine levels are closely related to motivation.

Response: Thank you very much for your valuable suggestion. We are now planning future studies targeting the impact of large-scale and long-term treatment of αGPC on human emotions and feelings. In addition, we have included a description of the factors affecting subjective feelings in the “Discussion” section of the revised manuscript (Lines 199-203).

In the results, it is reported that administration of αGPC increase motivation compared with the placebo group, but the p value is higher than threshold (p=0.0707), therefore that cannot be a significant outcome. Nevertheless, the results are supported by a weak statistic in general. Basically, there is not sufficient strong effects by administering of αGPC, as a significant effect only at nighttime point is observed, in figure 3D. Furthermore, also increases in the feeling of relief in both placebo and treatment groups are unclear (Fig. 3A).

Response: Thank you very much for your comments. We have revised these descriptions in the “Results” section of the revised manuscript (Lines 121-123, 119-120).

Overall graphs are clear and seem to be of good quality. It is advisable a general review of bibliography to update older references.

Response: We have updated the “References” section with recent publications.

Reviewer 3 Report

Dear authors,

The research paper entitled Alpha-glycerylphosphorylcholine increases motivation in 2 healthy volunteers: A single-blind, randomized, placebo-3 controlled human study is a randomised case study with placebo control group that aims to investigate the influence of Alpha-glycerylphosphorylcholine (αGPC) on human motivational processes.
The study initially seems interesting but suffers from a number of problems that imply major changes.
The first major problem is that a scale called KOKORO is used, of which neither its validity nor its reliability as a measurement tool is indicated. The second is that using only a single self-report test to quantify change is not sufficient to obtain valid results. It is necessary to measure with other instruments beyond self-report tests.
The sample size is very small, which could also be affecting the validity of the results. The minimum sample size in this type of study is about 30 subjects per group.
The paper does not study whether the use of the assessment tool (mobile application) is what is mediating motivation, so the results could be due to the use of this mobile application and not to the medication.
The introduction of the paper is shorter than the abstract and therefore does not fulfil the function of introducing the reader to the problem. There are no references on how to effectively measure motivation, for example, the concept of motivation is mixed up with that of emotion and anxiety, etc. It is not a well-conducted introduction.
Both the amounts of drug and the duration of treatment are not justified in the method section, it is wrongly done in the discussion.

For all these shortcomings, I consider that the work should not be published or that if it is published, all the relevant aspects mentioned above should be improved. With so many flaws it is impossible for the results to be valid and for any conclusions to be drawn.

Kind regards

Author Response

Thank you for your kind comments. Please see our replies to your comments below.

The study initially seems interesting but suffers from a number of problems that imply major changes.

The first major problem is that a scale called KOKORO is used, of which neither its validity nor its reliability as a measurement tool is indicated. The second is that using only a single self-report test to quantify change is not sufficient to obtain valid results. It is necessary to measure with other instruments beyond self-report tests.

Response: Thank you very much for your valuable suggestion. Our previous study validated that changes in autonomic nervous functions correlate with alterations in human feelings and emotions, using the KOKORO scale (Kume et al., Frontiers in Neuroscience 11 108 2017). Moreover, our collaborators have confirmed that inputting data related to human feelings with a smartphone well correlated with those using a paper-based visual analog scale (VAS) when a comparative test was performed by 20 human subjects.

The sample size is very small, which could also be affecting the validity of the results. The minimum sample size in this type of study is about 30 subjects per group.

Response: Thank you very much for your valuable suggestion. We are now planning further studies investigating the effects of large-scale and long-term treatment of αGPC, which will resolve this problem.

The paper does not study whether the use of the assessment tool (mobile application) is what is mediating motivation, so the results could be due to the use of this mobile application and not to the medication.

Response: Thank you very much for your valuable suggestion. We were certain that the increased motivation observed in this study was due to treatment with αGPC and not because of the use of the application since both placebo and αGPC treatment groups had input their feelings in the smartphone application under the same conditions.

The introduction of the paper is shorter than the abstract and therefore does not fulfil the function of introducing the reader to the problem. There are no references on how to effectively measure motivation, for example, the concept of motivation is mixed up with that of emotion and anxiety, etc. It is not a well-conducted introduction.

Response: Thank you very much for your kind suggestion. We have added text and revised the “Introduction” section in the manuscript (Line36-53).

Both the amounts of drug and the duration of treatment are not justified in the method section, it is wrongly done in the discussion.

Response: We have moved the description from the “Discussion” to the “Methods” section of the revised manuscript (Line 84-90).

Reviewer 4 Report

This is an interesting study on the role of alpha-GPC in increasing motivation in healthy subjects.

I have the following comments:

1) Your sample size was skewed in favor of females. Forty volunteers (31 females, 9 males). What were the plausible reasons for this skewed selection?

2) Were the subjects offered an incentive to participate in the study?

3) Is there a particular reason that you selected bedtime as the time of administration of Alpha-GPC?

4) On Page 3, you note “αGPC group was observed increased motivation compared with 98 the placebo group during the intervention period (p = 0.0707).”- The p-value is greater than 0.05 and thus statistically insignificant.

5) There are grammatical and syntax errors that need correction in the above sentence and throughout the paper.

6) It is not clear what is meant by morning, afternoon, and night? Was there a defined time at which the scales were completed in the morning, afternoon, and night? It would be helpful to have the specific times in the paper as night is a generic term( Eg. Does it mean between 8-12 pm or 8-10 pm. Some clarification with times would be helpful.

7) Page 6, “However, we showed that αGPC exhibited a tendency 147 to increase motivation during 1st week and had an effect on motivation during 2nd week 148 of intervention period” This sentence needs to be rewritten and clarified.

8) I am concerned about the quality of the discussion and the discussion needs to be elaborated further with additional references.

9) The paper requires edits throughout to clarify the methodology and to improve the scientific soundness.

Author Response

Thank you for your kind comments. Please see our replies to your comments below.

1) Your sample size was skewed in favor of females. Forty volunteers (31 females, 9 males). What were the plausible reasons for this skewed selection?

Response: We used flyers and an Internet website for recruiting research participants. Thus, we believe that women may be more interested in supplementation and improving human health compared to men.

.

2) Were the subjects offered an incentive to participate in the study?

Response:  We had provided the participants a small sum of money.

3) Is there a particular reason that you selected bedtime as the time of administration of Alpha-GPC?

Response: We selected bedtime for the administration of alpha-GPC since we believed that it is the best time to prevent the subjects from forgetting to take their medication.

4) On Page 3, you note “αGPC group was observed increased motivation compared with the placebo group during the intervention period (p = 0.0707).”- The p-value is greater than 0.05 and thus statistically insignificant.

Response: Thank you very much for your kind comment. We have revised this sentence in the “Results” section of the revised manuscript (Line121-123).

5) There are grammatical and syntax errors that need correction in the above sentence and throughout the paper.

Response: We have submitted the revised manuscript for a thorough English grammar and language check.

6) It is not clear what is meant by morning, afternoon, and night? Was there a defined time at which the scales were completed in the morning, afternoon, and night? It would be helpful to have the specific times in the paper as night is a generic term (Eg. Does it mean between 8-12 pm or 8-10 pm. Some clarification with times would be helpful.

Response: Thank you very much for your comments. We have described the specific time of each time zone (morning, afternoon, and night) in the original manuscript (Line 85-86) and the revised manuscript (Line 106-107).

7) Page 6, “However, we showed that αGPC exhibited a tendency to increase motivation during 1st week and had an effect on motivation during 2nd week of intervention period” This sentence needs to be rewritten and clarified.

Response: We have rewritten and clarified this sentence in the revised manuscript (Line 189-191).

8) I am concerned about the quality of the discussion and the discussion needs to be elaborated further with additional references.

Response: We have elaborated the discussion and added some references in the “Discussion” section of the revised manuscript.

9) The paper requires edits throughout to clarify the methodology and to improve the scientific soundness.

Response: We have submitted the revised manuscript for a thorough English grammar and language check.

Reviewer 5 Report

It is quite an interesting and helpful manuscript in which the authors investigated changes in subjective feelings of healthy volunteers using the KOKORO scale before and after the administration of αGPC. A total of 39 volunteers took part in a single blind, placebo-controlled project. Obtained data suggest that αGPC can be used to increase motivation in healthy individuals.

Although it is a valuable work having an interesting idea it needs some adjustment that I list below.

Some minor point:

In the part entitled Conclusion, besides the conclusion from presented data should be included a future perspective that would provide the reader with better understanding of scientific and general value of presented research. It should be added and then the paper could be accepted for publication.

I recommend publication after minor revision.

Author Response

Thank you for your kind comments. Please see our replies to your comments below.

In the part entitled Conclusion, besides the conclusion from presented data should be included a future perspective that would provide the reader with better understanding of scientific and general value of presented research. It should be added and then the paper could be accepted for publication.

Response: We have added sentences related to the future scope of work and broader implications of our research in the “Conclusion” section of the revised manuscript (Line 206-208).

Round 2

Reviewer 1 Report

Second review for Nutrients, Manuscript ID: nutrients-1205289: Alpha-glycerylphosphorylcholine increases motivation in healthy volunteers: A single-blind, randomized, placebo-controlled human study

The authors have not addressed issues 1, 2, and 3 that I raised in my previous review, which makes me wonder if the authors understand the related principles of evidence-based medicine. Here are those issues once again.

  1. No correction for multiple comparisons has been performed and no explanation provided for this omission. When performing 12 tests in a study, assuming independence of outcomes (although the 12 outcomes here would not all be independent) in approximately half of cases one would expect to find at least one outcome with p < 0.05. Thus, without correction for multiple comparisons it would be easy to “find” the existence of relationships among practically any unrelated variables.
  2. The explanation that the investigation and data analyses were performed by different researchers in no way addresses the problem of experimenter bias; and the explanation that this is a pilot study does not explain why the researchers chose a single-blind rather than double-blind format. I gave an example of evidence of this bias in the writing of the paper itself; and the revised paper presents more evidence of a bias toward presenting αGPC in a positive light: In the results section the poorer performance of αGPC concerning relief from anxiety was described as an association, while in the abstract the increase in motivation for αGPC (compared with placebo) was described as a causal relationship (“αGPC treatment increased motivation …”).
  3. The temporal relationship between taking the medication and changes in motivation and relief is not explored for this current study. It seems that the authors did not consider it important to know whether participants were rating their motivation before or after taking αGPC in the evening/night. Given that the nighttime increase in motivation is the only statistically significant finding after correcting for multiple comparisons, this seems like an important question.

Author Response

Reply to reviewer #1

Thank you for your comments. Please see our replies to your comments below.

  1. No correction for multiple comparisons has been performed and no explanation provided for this omission. When performing 12 tests in a study, assuming independence of outcomes (although the 12 outcomes here would not all be independent) in approximately half of cases one would expect to find at least one outcome with p < 0.05. Thus, without correction for multiple comparisons it would be easy to “find” the existence of relationships among practically any unrelated variables.

Response: Thank you very much for your valuable and patient suggestion. According to reviewer’s suggestion, we have performed re-analysis using Mann-Whitney U-test with Bonferroni multiple comparison corrections, and have revised these in the revised manuscript (Line 23-25, 110-112, 120-123, 160-162, 183-184).

  1. The explanation that the investigation and data analyses were performed by different researchers in no way addresses the problem of experimenter bias; and the explanation that this is a pilot study does not explain why the researchers chose a single-blind rather than double-blind format. I gave an example of evidence of this bias in the writing of the paper itself; and the revised paper presents more evidence of a bias toward presenting αGPC in a positive light: In the results section the poorer performance of αGPC concerning relief from anxiety was described as an association, while in the abstract the increase in motivation for αGPC (compared with placebo) was described as a causal relationship (“αGPC treatment increased motivation …”).

Response: We have re-analyzed the data using Bonferroni multiple comparison corrections as mentioned above, and carefully re-written those results (Line 110-112, 120-123).

In this study, actually, 32 out of 39 participants were subjected to double-blind study, while remaining 7 participants were subjected to single-blind study, because we needed to provide test capsules to all participants for a few days in the short schedule of study. For the reason, this study was assessed as a single-blind study in all data from 39 participants. Moreover, we tried to analyze only the data which had been obtained from 32 participants under double-blind manner. The results showed the similar conclusion to that in all data from 39 participants.

  1. The temporal relationship between taking the medication and changes in motivation and relief is not explored for this current study. It seems that the authors did not consider it important to know whether participants were rating their motivation before or after taking αGPC in the evening/night. Given that the nighttime increase in motivation is the only statistically significant finding after correcting for multiple comparisons, this seems like an important question.

Response: We instructed all participants to input their feeling data every night before taking test capsules to avoid a short-term effect including placebo effect. We have added the description about that in “Methods” section in the revised manuscript (Line 105).

Once again, thank you very much for your generous comments.

Reviewer 2 Report

The authors give a response to my questions. I think that the paper is acceptable in the present form

Author Response

Thank you very much for your valuable and patient suggestion.

Reviewer 3 Report

Dear authors,

Your paper has been improved, yet it still has methodological problems that prevent it from being used as a meta-analysis study. I have accepted it for publication because I believe that its results, although very limited, can serve as a pilot study.

Cordial greeetings

Author Response

(The authors gave the same response as above.)

Reviewer 4 Report

Thanks for making the requested edits. Though there are still issues related to robust scientific soundness, the paper is overall acceptable in its present form.

Author Response

(The authors gave the same response as above.)
